analytical chemistry

length-based detection, thread-based analytical device, polyphenols, instrument-free

**Author for correspondence:**
Longfei Cai
e-mail: cailf@hstc.edu.cn

†These authors contributed equally.
This article has been edited by the Royal Society of Chemistry, including the commissioning, peer review process and editorial aspects up to the point of acceptance.

# Instrument-free detection of polyphenols with a thread-based analytical device

Jiahong Song†, Zhuang Ouyang†, Wei Lu
and Longfei Cai

School of Chemistry and Environmental Engineering, Hanshan Normal University, Chaozhou, Guangdong 521041, People's Republic of China

(iD) LC, 0000-0002-4959-8723

We described an instrument-free method for quantitative analysis of the total content of tea polyphenols by measurement of the length of a coloured band. Polyphenols react with ferrous ions to form a colourless ferrous-polyphenols complex on cotton threads, which could be adsorbed on the threads. The complex was then oxidized to form a blue-black ferric-polyphenols complex, generating a blue-black band on the cotton thread. The length of this blue-black band was then measured to detect the total content of polyphenols. The advantages of this method include low cost, rapid analysis, low consumption, easy fabrication and operation. Furthermore, the digital instrument (scanner or camera) as well as the image processing software are not required. This proposed method was used to detect polyphenols in tea leaf extracts with an analytical result agreeing well with that obtained by a standard method, which demonstrates its potential in monitoring of tea leaf quality, especially in resource-limited regions and settings.

## 1. Introduction

Polyphenols are one of the main constituents in tea leaves. Various functions of tea polyphenols have received much attention, including antioxidant, antiradical and anti-thrombotic activities. The total content of polyphenols is an important factor that may influence the quality of tea leaf. Therefore, a facile and rapid method for tea polyphenols assay with low cost is highly desirable.

Thus far, the total content of tea polyphenols has been quantified with various analytical methods such as visible spectrophotometry [1–3], atomic absorption spectrometry [4] and chromatography [5]. Among these instrument-based methods, spectrophotometric methods are most commonly used. For example, ferrous tartrate was used as a chromogenic reagent in a national standard method for quantitative analysis of tea polyphenols [6]. Based on the formation of a ternary

bluish-violet complex owing to the reaction that occurred among ferrous ion, tartrate and polyphenols, tea polyphenols could be quantified by measuring the absorbance of the bluish-violet complex. However, these above-mentioned methods are limited by the expensive instruments, time-consuming operations, large sample/reagents consumption and requirement of trained personnel. Microfluidic paper-based analytical device (µPAD) is featured as low cost, reduced sample/reagent consumption, rapid analysis speed and easy operation. Since this technique was developed by Whitesides group in 2007 [7], µPAD has been widely used as a cost-effective platform for performing chemical and biochemical analysis, for example, food testing [8,9], environmental analysis [10,11] and clinical diagnostics [12,13]. In 2016, Hao et al. fabricated µPADs with a flower-shaped pattern to detect polyphenols [14]. This method is based on the chromogenic reaction that occurred between Folin–Ciocalteu phenol reagent and tea polyphenols on the detection zones. After the image of the detection zones was taken by a scanner, the intensity was extracted using an image processing software to detect polyphenols in tea leaf extracts. This method is simple and straightforward for detection of the total content of tea polyphenols. However, an electronic instrument such as a scanner or digital camera was required. Additionally, when using an image processing software to obtain colour intensities, the intensity value is correlated with reaction time, humidity and lighting conditions when the images were captured. These limitations pose difficulties for accurate quantification of the total content of tea polyphenols.

Cotton thread is another cheap alternative for fabricating microfluidic analytical devices. The features of cotton thread include light weight, flexibility, disposability and it is difficult to break. Furthermore, the aqueous liquid could flow on the cotton thread owing to the hydrophilic property of thread, thereby eliminating the use of external fluid driving apparatus. Additionally, a number of detection techniques could also be coupled with microfluidic thread-based analytical devices (µTAD) for quantitative analysis, including colorimetry [15–17], fluorometry [18], electrochemistry [19,20] and chemiluminescence [21]. Benefiting from these advantages, the thread has been widely used for developing cost-effective analytical platforms for chemical and biochemical assays since µTAD was described by Li et al. and Reches et al. in 2010 [22,23]. To the best of our knowledge, however, µTAD has not been used to detect tea polyphenols so far. In this work, we fabricated a simple µTAD for rapid detection of the total content of tea polyphenols with a ruler. Tea polyphenols react with ferrous ions to produce a colourless complex, which could be deposited onto the thread owing to the adsorption of complex onto the thread. The colourless complex was then oxidized to be a blue-black complex, forming a blue-black band on the cotton thread. Quantification of the total content of tea polyphenols was achieved by measuring the coloured band length. This method is free of any electronic instruments, which is very suitable for detection of tea polyphenols and monitoring of tea leaf quality in resource-limited regions and settings.

# 2. Material and methods

## 2.1. Reagents and apparatus

All reagents used were of analytical grade unless stated otherwise. Tea polyphenols were purchased from Jianglai Biotechnology Co., Ltd, Shanghai, China. Sulfate heptahydrate and sodium carbonate were purchased from Xilong Chemical Co., Ltd, Shantou, China. To obtain $2.0 \, g \, l^{-1}$ of standard stock tea polyphenols solution, $0.20 \, g$ of tea polyphenols was dissolved in water followed by dilution to 100 ml with water. The standard working tea polyphenols solutions with varying concentrations were prepared by serial dilution of the stock standard solution. Ferrous sulfate heptahydrate prepared in 1 : 1 ethanol/water was used as chromogenic reagents. After the solution was prepared, $0.2 \, g$ of iron powder was added into the solution to prevent $Fe^{2+}$ from being oxidized. Sodium carbonate solution ($10.0 \, g \, l^{-1}$) was prepared with water. Ultrapure water ($18.25 \, M\Omega \, cm$) produced with a water purification system (EPED-EQ-10T, Nanjing Eped Technology Development Co., Ltd, Nanjing, China) was used throughout. Cotton threads purchased from the local commercial market were used to fabricate the thread-based devices.

## 2.2. Fabrication of thread-based analytical device

The cotton threads obtained from a local store are hydrophobic owing to the natural hydrophobic substances present in the thread fibres. Thus, the threads should be modified to be hydrophilic before

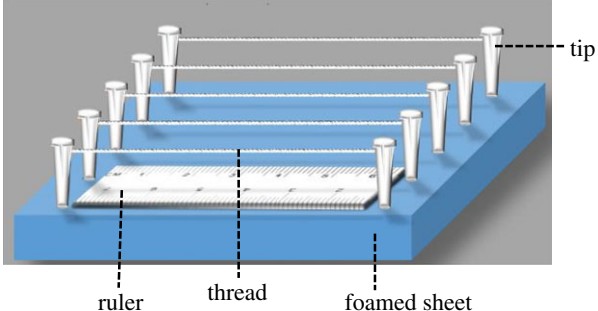

**Figure 1.** Schematic diagram of a thread-based analytical device.

fabricating thread-based devices. Sodium carbonate solution was used to remove the hydrophobic substances in threads. Specifically, the cotton threads were firstly soaked in 10 g l$^{-1}$ Na$_2$CO$_3$ solution at 100°C for 1 h, and then, the cotton threads were washed with water until sodium carbonate in the threads was removed. The thread-based device was fabricated with micropipette tips, a knife, a foamed sheet (120 × 120 mm) and 0.4 mm diameter cotton threads (100 mm long) as described in [24]. Briefly, slits were made on the tips followed by inserting into a foamed sheet. The thread was fixed onto the tips by fixing two ends of the thread into tip slits (figure 1).

## 2.3. Sample pretreatment

Dried tea leaves (1.0 g) were accurately weighed into a 100 ml beaker, followed by pouring 70 ml of boiling water into the beaker. A boiling water bath was used to extract polyphenols in tea leaves for 20 min. The mixtures were filtered to obtain a filtrate. This filtrate was diluted to 100 ml, which was used as a sample.

## 2.4. Procedure for tea polyphenols assay

Ferrous sulfate solution (5.0 μl) was pipetted onto the cotton thread with a pipette, which allows the liquid to wick along the thread. The thread was then allowed to air-dry for 2.5 min followed by dropping 7.5 μl of tea polyphenols solution onto the thread. After being air-dried for 8 min, a blue-black band was formed on the thread. The length of the blue-black band was then measured for quantitative analysis of the total content of tea polyphenols.

# 3. Results and discussion

## 3.1. Principle

The length-based detection technique on threads usually relies on the adsorption or precipitation of coloured reaction product on threads [25]. In this work, the hydrophilic cotton threads were firstly modified with a chromogenic reagent (ferrous sulfate solution) by dropping chromogenic reagents onto threads (figure 2a). As sample or standard solutions containing tea polyphenols were spotted onto the thread which was previously modified with the chromogenic reagents, a colourless ferrous-tea polyphenols complex on the thread was formed due to the reaction that occurred between ferrous ions and tea polyphenols (figure 2b). The complex was adsorbed and deposited onto thread fibres until all the analytes were consumed. As the devices were allowed to air-dry, the colourless ferrous-polyphenols could be oxidized to produce a blue-black ferric-polyphenols complex. Thus, a blue-black band was formed on the cotton thread (figure 2c). After the linear correlation between the band length and contents of tea polyphenols was plotted, the total contents of tea polyphenols in the sample were quantified by measurement of the band length that resulted from the sample solution.

## 3.2. Selection of chromogenic reagents

Ferrous tartrate [6,26] and Folin–Ciocalteu phenol reagent [26,27] are the most commonly used chromogenic reagents for determination of polyphenols. Both ferrous tartrate-based and Folin–Ciocalteu-based

(a)

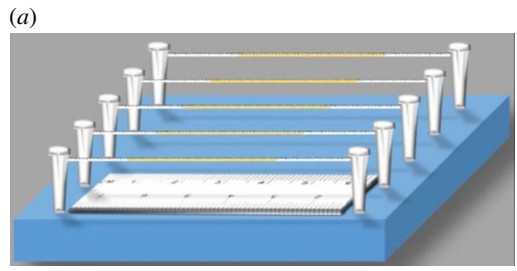

(b)

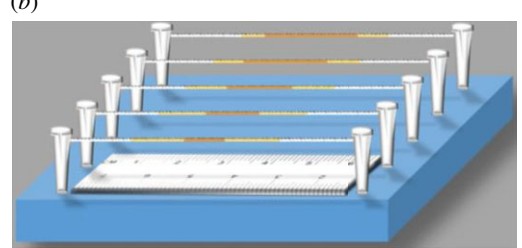

(c)

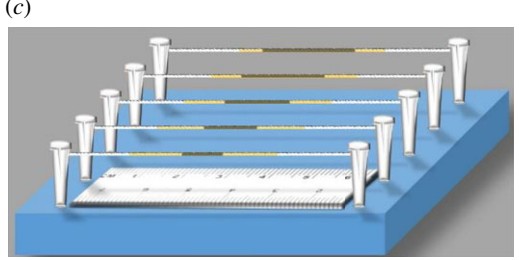

**Figure 2.** Schematic diagrams illustrating deposition of chromogenic reagents (a), formation of a ferrous-polyphenols complex (b) and formation of blue-black bands (c) on cotton threads.

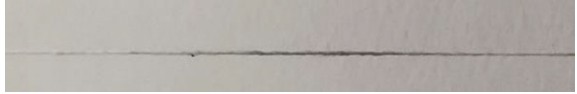

**Figure 3.** Image obtained after adding tea polyphenols solution onto the cotton thread modified with ferrous tartrate. Concentration of tea polyphenols: 0.4 g l$^{-1}$; concentration of Fe$^{2+}$: 0.4 g l$^{-1}$; concentration of tartrate: 6.2 g l$^{-1}$; volume of Fe$^{2+}$ solution: 5.0 µl; and volume of tea polyphenols solution: 7.5 µl.

spectrophotometric methods have been used as national standards for detection of polyphenols in tea leaf or tea leaf extracts. Both of these two chromogenic reagents could react with tea polyphenols to produce a coloured complex on cotton threads. However, the boundary of the coloured band on the cotton thread is not clear (figure 3), which makes measurement of the band length difficult. Furthermore, the band length varied little with the contents of tea polyphenols when using ferrous tartrate or Folin–Ciocalteu phenol reagent as chromogenic reagents. These issues may be due to the fact that the coloured product could not be adsorbed by the thread fibre. Thus, both ferrous tartrate and Folin–Ciocalteu phenol reagent are not suitable as chromogenic reagents for the length-based detection of the total content of tea polyphenols on cotton threads.

## 3.3. Effect of Fe$^{2+}$ concentration

Fe$^{2+}$ (0.2–0.8 g l$^{-1}$) was prepared for studying the effect of chromogenic reagent concentration on the length and intensity of the coloured band (tea polyphenols concentration was kept constant at 0.4 g l$^{-1}$). As the concentration of Fe$^{2+}$ increased, the intensity of the coloured band increased while the blue-black band length decreased with the concentration of Fe$^{2+}$ (figure 4). Fe$^{2+}$ (0.4 g l$^{-1}$) was used in this work by compromising the colour intensity and the length of the blue-black band formed on the cotton threads.

## 3.4. Effect of pH

The effect of pH of chromogenic reagents in the range of 1.0–6.0 on the formation of the coloured band was studied by keeping the concentration of Fe$^{2+}$ and tea polyphenols constant at 0.4 g l$^{-1}$ (figure 5). The results indicated that Fe$^{2+}$ and tea polyphenols could not react to generate the coloured complex on the cotton threads with ferrous sulfate solution of pH 1.0. The intensity of the coloured band increased with pH, while the band length decreased with pH in the range of 2–3. Meanwhile, the reaction time needed for the formation of the coloured band decreased from 15 to 7 min when pH

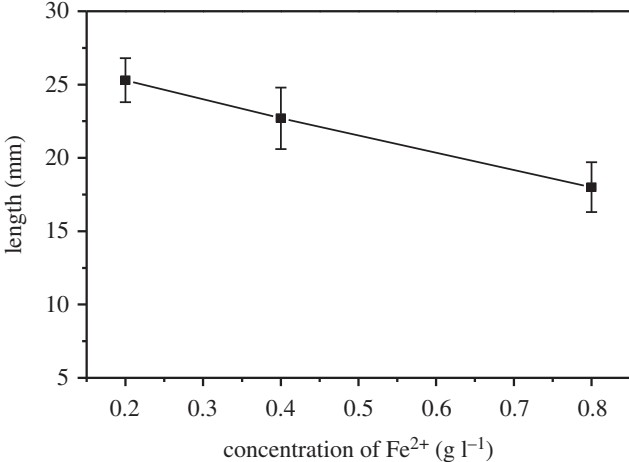

**Figure 4.** The length of the blue-black band obtained by varying concentration of $Fe^{2+}$. The error bars were obtained from three replica runs. Concentration of tea polyphenols: 0.4 g l$^{-1}$; pH of $Fe^{2+}$ solution: 4.5; volume of $Fe^{2+}$ solution: 5.0 μl; and volume of tea polyphenols solution: 7.5 μl.

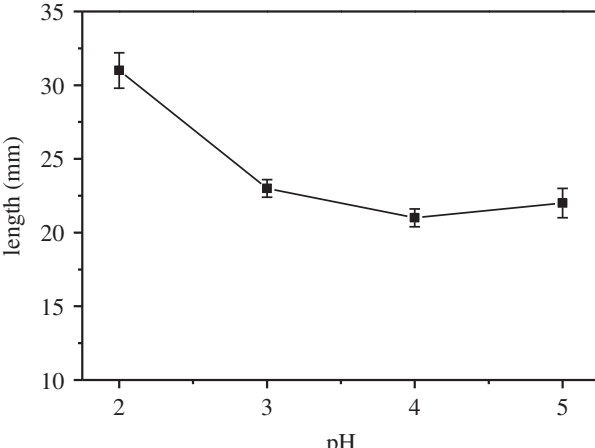

**Figure 5.** The length of the coloured band varying with pH of the chromogenic reagent solution. The error bars were obtained from three replica runs. Concentration of $Fe^{2+}$: 0.4 g l$^{-1}$. Other conditions were the same as those in figure 4.

was increased from 2 to 3. Both the intensity and band length varied little with pH in the range of 3–5. At a pH higher than 5.5, precipitates form in the chromogenic reagent solutions; this may be due to the hydrolysis of ferrous ions. Therefore, pH in the range of 3–5 is suitable for analytical applications by compromising the colour intensity and reaction time. In this work, a sulfate solution of pH 4.5 was used for analytical applications, which was prepared by dissolving ferrous sulfate in 1 : 1 ethanol/water without the addition of any acid or base solution.

## 3.5. Effect of foreign species

The effect of main constituents contained in tea leaves on determination of 0.4 g l$^{-1}$ tea polyphenols was studied, including sugar, ascorbic acid, proteins, amino acids and caffeine. To study the effect of these constituents on determination of tea polyphenols, two solutions were prepared. One contains 0.4 g l$^{-1}$ tea polyphenols and the other contains 0.4 g l$^{-1}$ tea polyphenols as well as a foreign species with certain contents. The two solutions were added onto individual threads modified with chromogenic reagents, the lengths of blue-black bands generated by the two solutions were then measured and compared to evaluate the interference of foreign species. The tolerance level was defined as the concentration of foreign species that caused a length variation lower than 5%. The tolerance levels for determination of 0.4 g l$^{-1}$ tea polyphenols are 0.4, 0.06, 0.4, 0.2 and 0.4 g l$^{-1}$ for sugar, ascorbic acid, protein, glutamic acid and caffeine, respectively.

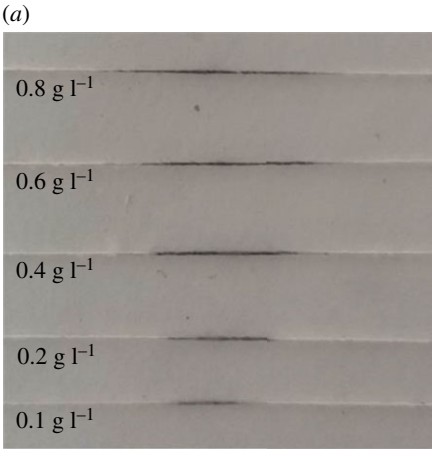
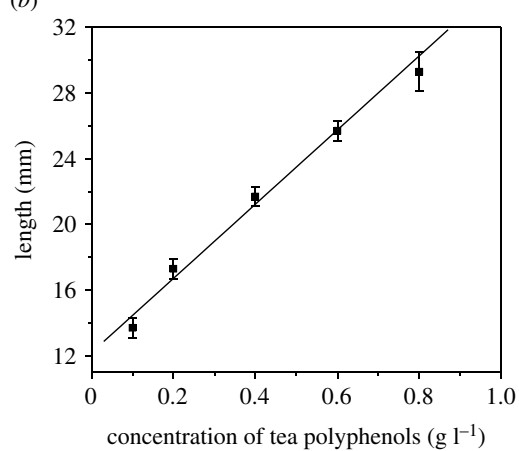

**Figure 6.** The effect of tea polyphenols concentration on the length of blue-black band. (*a*) An image of coloured bands formed on cotton threads with varying concentration of tea polyphenols. (*b*) Calibration curve obtained by varying tea polyphenols concentration in the range of 0.1–0.8 g l$^{-1}$. The error bars were obtained from three replica runs. Concentration of Fe$^{2+}$: 0.4 g l$^{-1}$. Other conditions were the same as those in figure 4.

## 3.6. Linearity range and limits of detection for tea polyphenols

Tea polyphenols solutions with varying concentrations (0.1–0.8 g l$^{-1}$) were prepared to plot a linear correlation under the optimal conditions described above. The linear correlation was then used to quantify tea polyphenols in samples. After the standard tea polyphenols solutions were added onto threads modified with Fe$^{2+}$, the coloured bands were formed (figure 6*a*) and the lengths of these bands were measured. The linear relation between the coloured band length (*L*, mm) and tea polyphenols concentration (*C*, g l$^{-1}$) was *L* (mm) = 22.6*C* (g l$^{-1}$) + 12.2 (figure 6*b*). The correlation coefficient of this curve is 0.994. To obtain the detection limit of tea polyphenols using this proposed method, a solution containing 0.05 g l$^{-1}$ tea polyphenols was measured 11 times. The standard deviation was calculated as 0.7 mm. The limit of detection of tea polyphenols is thus 0.09 g l$^{-1}$ according to the equation of 3*S*/*K*.

## 3.7. Real sample analysis

The total content of polyphenols in tea leaf extracts was detected with this proposed thread-based device. The tea leaves were pretreated as described in the experimental section. The obtained sample solution was further diluted by adding 250 µl of the sample into a 1.5 ml centrifuge tube followed by diluting to 1.0 ml. Detection of tea polyphenols in the diluted sample was then performed on the thread-based devices as described above. The average length of the blue-black band is 21.7 ± 1.5 mm (*n* = 3). The total content of tea polyphenols was 0.42 ± 0.06 g l$^{-1}$ in the diluted tea leaf extracts. Therefore, the total content in the tea leaf extracts was calculated as 1.68 ± 0.24 g l$^{-1}$. Considering that the total content of polyphenols is usually much higher than other common constituents in tea leaf extracts, the diluted sample was analysed without any further pretreatment. The accuracy of this proposed method was evaluated with a ferrous tartrate-based standard method [6]. The measured total content of polyphenols in the tea leaf extracts is 1.80 + 0.04 g l$^{-1}$, which compared well with that obtained by this proposed thread-based method (1.68 ± 0.24 g l$^{-1}$).

## 4. Conclusion

We developed a facile method for determination of the total content of tea polyphenols in tea leaf extracts on cotton threads with minimum cost. Quantitative analysis of tea polyphenols was achieved by measurement of the band length formed on the threads. This simple, portable and cost-effective method could be used to detect total content of tea polyphenols with rapid analysis. Furthermore, no electronic instrument was required during quantitative analysis. We believe that these features make this method very promising and suitable for tea leaf quality control and monitoring especially in resource-limited setting and regions.

Data accessibility. The datasets supporting this article have been uploaded as part of the supplementary material.

Authors' contributions. J.S. and Z.O. fabricated the thread-based devices and used them to detect polyphenols. W.L. evaluated the accuracy of the analytical results using a standard method. L.C. supervised the experiment and wrote the paper.

Competing interests. We declare we have no competing interests.

Funding. This research was funded by the Guangdong Provincial Education Research Innovation Fund (no. 2018GXJK108), Guangdong Provincial Science and Technology Innovation Fund for Undergraduates (pdjh2019b0314) and Guangdong Provincial Higher Education Research and Reform Fund.

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
