## [Reviewer comments · Royal Society Open Science]

Review History

RSOS-192130.R0 (Original submission)

Review form: Reviewer 1

Is the manuscript scientifically sound in its present form?

Yes

Are the interpretations and conclusions justified by the results?

Yes

Is the language acceptable?

Yes

Do you have any ethical concerns with this paper?

No

Have you any concerns about statistical analyses in this paper?

No

Recommendation?

Accept with minor revision (please list in comments)

Comments to the Author(s)

This study presented a novel and cost-effective thread-based method for determination of polyphenols in tea leaf extracts based on length measurements of color bands formed in cotton threads between tea polyphenols and ferrous ions. This analytical method is free of external expensive instruments, which is very interesting and promising for resource-limited regions and settings. I recommend its publication in this journal after fully considering several minor issues as follows:

1. Tea polyphenols are a class of phenols found in green tea including epigallocatechin gallate (EGCG), epigallocatechin, epicatechin gallate, and epicatechin; flavanols such as kaempferol, quercetin, and myricitin. This study dedicates to total content of tea polyphenols rather than specific or each constitute. Therefore please use total polyphenol content or total content of tea polyphenols all over the text.
2. Colorless ferrous-polyphenol complexes were oxidized to blue-black complexes by oxygen gas in the air or the solvents. I am worrying about variation in the oxidation degree induced naturally. Would you please check if the oxidation was accomplished by using such external oxidants as hydrogen peroxide, sodium hypochlorite, etc.
3. In the estimation of interference from coexisting substances, amino acids were considered. As amino acids are a large class of biological molecules, do you test which type of amino acids? If so, please specify the used amino acids.
4. Please provide detailed dimensions of the thread-based analytical device.
5. The description regarding the pretreatment of tea leaves should be placed in the experimental supporting by literature.
6. The manuscript should be proofread to avoid grammatical errors. Some typical grammatical errors are marked in the attached pdf (Appendix A).

Review form: Reviewer 2

Is the manuscript scientifically sound in its present form?

Yes

Are the interpretations and conclusions justified by the results?

Yes

Is the language acceptable?

Yes

Do you have any ethical concerns with this paper?

No

Have you any concerns about statistical analyses in this paper?

No

Recommendation?

Accept with minor revision (please list in comments)

Comments to the Author(s)

In the manuscript (RSOS-192130), Song et al. developed an instrument-free method for polyphenol assay with cotton threads. Quantitative analysis of tea polyphenols was achieved by measurement of band length formed on the threads. It is simple, portable and cost-effective with rapid analysis speed. The results are interesting and it should be of broad interest to the readership of RSOS. But some parts of the text need more detailed explanation and correction. So I recommend this manuscript for publication after minor revisions towards the following points:

1. Ferrous sulphate is soluble in aqueous liquid, why did the authors prepare ferrous sulphate solutions with 1:1 ethanol/water?
2. Why the author does not use the untreated cotton threads to fabricate the device? Are there any other method or solution could be used to pretreat the hydrophobic threads besides Na₂CO₃ solution?
3. In the manuscript, author claimed that the limit of detection of tea polyphenols is thus 0.09 g L⁻¹ however in the text, "a solution containing 0.05 g L⁻¹ tea polyphenols was measured 11 times ..." How can such a low concentration of signals be detected?
4. Could distance-based detection of polyphenols be performed and achieved on paper-based analytical device using reaction between ferrous sulphate and polyphenols?

Decision letter (RSOS-192130.R0)

30-Jan-2020

Dear Dr Cai:

Title: Instrument-Free Detection of Polyphenols with a Thread-Based Analytical Device
Manuscript ID: RSOS-192130

Thank you for submitting the above manuscript to Royal Society Open Science. On behalf of the Editors and the Royal Society of Chemistry, I am pleased to inform you that your manuscript will be accepted for publication in Royal Society Open Science subject to minor revision in accordance with the referee suggestions. Please find the reviewers' comments at the end of this email.

The reviewers and handling editors have recommended publication, but also suggest some minor revisions to your manuscript. Therefore, I invite you to respond to the comments and revise your manuscript.

Because the schedule for publication is very tight, it is a condition of publication that you submit the revised version of your manuscript before 08-Feb-2020. Please note that the revision deadline will expire at 00.00am on this date. If you do not think you will be able to meet this date please let me know immediately.

- 1) A text file of the manuscript (tex, txt, rtf, docx or doc), references, tables (including captions) and figure captions. Do not upload a PDF as your "Main Document".
- 2) A separate electronic file of each figure (EPS or print-quality PDF preferred (either format should be produced directly from original creation package), or original software format)

- 3) Included a 100 word media summary of your paper when requested at submission. Please ensure you have entered correct contact details (email, institution and telephone) in your user account
- 4) Included the raw data to support the claims made in your paper. You can either include your data as electronic supplementary material or upload to a repository and include the relevant doi within your manuscript
- 5) All supplementary materials accompanying an accepted article will be treated as in their final form. Note that the Royal Society will neither edit nor typeset supplementary material and it will be hosted as provided. Please ensure that the supplementary material includes the paper details where possible (authors, article title, journal name).

Best wishes,
Dr Laura Smith
Publishing Editor, Journals

RSC Associate Editor:
Comments to the Author:
(There are no comments.)

RSC Subject Editor:
Comments to the Author:
(There are no comments.)

Reviewer comments to Author:
Reviewer: 1

Comments to the Author(s)

This study presented a novel and cost-effective thread-based method for determination of polyphenols in tea leaf extracts based on length measurements of color bands formed in cotton threads between tea polyphenols and ferrous ions. This analytical method is free of external expensive instruments, which is very interesting and promising for resource-limited regions and

settings. I recommend its publication in this journal after fully considering several minor issues as follows:

1. Tea polyphenols are a class of phenols found in green tea including epigallocatechin gallate (EGCG), epigallocatechin, epicatechin gallate, and epicatechin; flavanols such as kaempferol, quercetin, and myricitin. This study dedicates to total content of tea polyphenols rather than specific or each constitute. Therefore please use total polyphenol content or total content of tea polyphenols all over the text.
2. Colorless ferrous-polyphenol complexes were oxidized to blue-black complexes by oxygen gas in the air or the solvents. I am worrying about variation in the oxidation degree induced naturally. Would you please check if the oxidation was accomplished by using such external oxidants as hydrogen peroxide, sodium hypochlorite, etc.
3. In the estimation of interference from coexisting substances, amino acids were considered. As amino acids are a large class of biological molecules, do you test which type of amino acids? If so, please specify the used amino acids.
4. Please provide detailed dimensions of the thread-based analytical device.
5. The description regarding the pretreatment of tea leaves should be placed in the experimental supporting by literature.
6. The manuscript should be proofread to avoid grammatical errors. Some typical grammatical errors are marked in the attached pdf.

Reviewer: 2

Comments to the Author(s)

In the manuscript (RSOS-192130), Song et al. developed an instrument-free method for polyphenol assay with cotton threads. Quantitative analysis of tea polyphenols was achieved by measurement of band length formed on the threads. It is simple, portable and cost-effective with rapid analysis speed. The results are interesting and it should be of broad interest to the readership of RSOS. But some parts of the text need more detailed explanation and correction. So I recommend this manuscript for publication after minor revisions towards the following points:

1. Ferrous sulphate is soluble in aqueous liquid, why did the authors prepare ferrous sulphate solutions with 1:1 ethanol/water?
2. Why the author does not use the untreated cotton threads to fabricate the device? Are there any other method or solution could be used to pretreat the hydrophobic threads besides Na₂CO₃ solution?
3. In the manuscript, author claimed that the limit of detection of tea polyphenols is thus 0.09 g L⁻¹. However in the text, "a solution containing 0.05 g L⁻¹ tea polyphenols was measured 11 times ..." How can such a low concentration of signals be detected?
4. Could distance-based detection of polyphenols be performed and achieved on paper-based analytical device using reaction between ferrous sulphate and polyphenols?

Author's Response to Decision Letter for (RSOS-192130.R0)

See Appendix B.

Decision letter (RSOS-192130.R1)

11-Feb-2020

Dear Dr Cai:

Title: Instrument-Free Detection of Polyphenols with a Thread-Based Analytical Device
Manuscript ID: RSOS-192130.R1

It is a pleasure to accept your manuscript in its current form for publication in Royal Society Open Science. The chemistry content of Royal Society Open Science is published in collaboration with the Royal Society of Chemistry.

RSC Associate Editor
Comments to the Author:
(There are no comments.)

Reviewer(s)' Comments to Author:

Appendix A**ROYAL SOCIETY
OPEN SCIENCE****Instrument-Free Detection of Polyphenols with a Thread-
Based Analytical Device**

Journal:	Royal Society Open Science
Manuscript ID	RSOS-192130
Article Type:	Research
Date Submitted by the Author:	18-Dec-2019
Complete List of Authors:	Song, Jiahong; Hanshan Normal University Ouyang, Zhuang; Hanshan Normal University Lu, Wei; Hanshan Normal University Cai, Longfei; Hanshan Normal University,
Subject:	Analytical chemistry < CHEMISTRY
Keywords:	length-based detection, thread-based analytical device, polyphenols, instrument-free
Subject Category:	Chemistry

Author-supplied statements

Relevant information will appear here if provided.

Ethics

Does your article include research that required ethical approval or permits?:

This article does not present research with ethical considerations

Statement (if applicable):

CUST_IF_YES_ETHICS :No data available.

Data

It is a condition of publication that data, code and materials supporting your paper are made publicly available. Does your paper present new data?:

Yes

Statement (if applicable):

it is provided as supplementary material

Conflict of interest

I/We declare we have no competing interests

Statement (if applicable):

CUST_STATE_CONFLICT :No data available.

Authors' contributions

This paper has multiple authors and our individual contributions were as below

Statement (if applicable):

Jiahong Song and Zhuang Ouyang fabricated the thread-based devices and used them to detect polyphenols. Wei Lu evaluate the accuracy of the analytical results using a standard method. Longfei Cai supervised the experiment and wrote the paper.

Instrument-free detection of polyphenols with a thread-based analytical device

Jiahong Song,[†] Zhuang Ouyang,[†] Wei Lu and Longfei Cai*

School of Chemistry and Environmental Engineering, Hanshan Normal University, Chaozhou, Guangdong 521041, China.

Keywords: Length-based detection, thread-based analytical device, polyphenols, instrument-free

1. Summary

We described an instrument-free method for quantitative analysis of tea polyphenols by measurement of the length of colored band. Polyphenols react with ferrous ions to form a colorless ferrous-polyphenols complex on cotton threads, which could be adsorbed on the threads. The complex was then oxidized to form a blue-black ferric-polyphenols complex, generating a blue-black band on the cotton thread. The length of this blue-black band was then measured to detect polyphenols. The advantages of this method include low cost, rapid analysis speed, low consumption, easy fabrication and operation. Furthermore, the digital instrument (scanner or camera) as well as image processing software are not required. This proposed method was used to detect polyphenols in tea leaf extracts with an analytical result agreeing well with that obtained by a standard method, which demonstrates its potentials in monitoring of tea leaf quality especially in those resource-limited regions and settings.

2. Introduction

Polyphenols are one of the main constituents in tea leaves. Various functions of tea polyphenols have received much attention, including antioxidant, antiradical and anti-thrombotic activities. Contents of polyphenols is an important factor that may influence the quality of tea leaf. Therefore, a facile and rapid method for tea polyphenols assay with low cost is highly desirable.

Thus far, tea polyphenols have been quantified with various analytical methods such as visible spectrophotometry [1-3], atomic absorption spectrometry [4] and chromatography [5]. Among these instrument-based methods, spectrophotometric methods are most commonly used. For example, ferrous tartrate was used as a chromogenic reagent in a national standard method for quantitative analysis of tea polyphenols [6]. Based on the formation of a ternary bluish violet complex owing to the reaction occurred between ferrous ion, tartrate and polyphenols, tea polyphenols could be quantified by measuring the absorbance of the bluish violet complex. However, these above-mentioned methods are limited by the expensive instruments, time-consuming operations, large sample/reagents consumption and requirement of trained personnel. Microfluidic paper-based analytical device (μ PAD) is featured as low cost, reduced sample/reagent consumption, rapid analysis speed and easy operation. Since this technique was developed by Whitesides group in 2007 [7], μ PAD has been widely used as a cost-effective platform for performing chemical and biochemical analysis, for example food testing [8,9], environmental analysis [10,11] and clinical diagnostics [12,13]. In 2016, Hao et al. fabricated μ PADs with a flower-shaped pattern to detect polyphenols [14]. This method is based on the chromogenic reaction occurred between Folin-Ciocalteu phenol reagent and tea polyphenols on the detection zones. After the image of the detection zones was taken by a scanner, the intensity was extracted using an image processing software to detect polyphenols in tea leaf extracts. This method is simple and straightforward for detection of tea polyphenols. However, electronic instrument such as a scanner or digital camera was required. Additionally, when using an image processing software to obtain color intensities, intensity value is correlated with reaction time, humidity and lightening conditions when the images was captured. These limitations pose difficulties for accurate quantification of tea polyphenols.

Cotton thread is another cheap alternative for fabricating microfluidic analytical devices. The features of cotton thread include lightweight, flexibility, disposability and it is difficult to break. Furthermore, aqueous liquid could flow on the cotton thread owing to the hydrophilic property of thread, thereby eliminating the use of external fluid driving apparatus. Additionally, a number of detection techniques could also be coupled with microfluidic thread-based analytical device (μ TAD) for quantitative analysis, including colorimetry [15-17], fluorometry [18], electrochemistry [19,20] and

*Author for correspondence (caifl@hstc.edu.cn).

[†]These authors contribute equally.

chemiluminescence [21]. Benefiting from these advantages, thread has been widely used for developing cost-effective analytical platforms for chemical and biochemical assays since μ TAD was described by Li et al. and Reches et al. in 2010 [22,23]. To the best of our knowledge, however, μ TAD has not been used to detect tea polyphenols thus far. In this work, we fabricated a simple μ TAD for rapid detection of tea polyphenols with a ruler. Tea polyphenols reacts with ferrous ions to produce colorless complex, which could be deposited onto thread owing to the adsorption of complex onto thread. The colorless complex was then oxidized to be a blue-black complex, forming a blue-black band on the cotton thread. Quantification of tea polyphenols was achieved by measuring the colored band length. This method is free of any electronic instruments, which is very suitable for detection of tea polyphenols and monitoring of tea leaf quality in those resource-limited regions and settings.

3. Materials and Methods

3.1 Reagents and Apparatus

All reagents used were of analytical grade unless stated otherwise. Tea polyphenols was purchased from Jianglai Biotechnology Co., Ltd., in Shanghai of China. Sulfate heptahydrate and sodium carbonate were purchased from Xilong Chemical Co., Ltd. in Shantou of China. 2.0 g L^{-1} of standard stock tea polyphenols solution was obtained by dissolving 0.20 g of tea polyphenols in water followed by dilution to 100 mL with water. The standard working tea polyphenols solutions with varying concentrations were prepared by serial dilution of the stock standard solution. Ferrous sulfate heptahydrate prepared in 1:1 ethanol/water was used as chromogenic reagents. After the solution was prepared, 0.2 g of iron powder was added into the solution to prevent Fe^{2+} from being oxidized. 10.0 g L^{-1} of sodium carbonate solution was prepared with water. Ultrapure water ($18.25 \text{ M}\Omega \text{ cm}$) produced with a water purification system (EPED-EQ-10T, Nanjing Eped Technology Development Co., Ltd, Nanjing, China) was used throughout. The cotton threads purchasing from the local commercial market was used to fabricate the thread-based devices.

3.2 Fabrication of Thread-based Analytical Device

The cotton threads obtained from a local store is hydrophobic owing to the natural hydrophobic substances present in the thread fibers. Thus, the threads should be modified to be hydrophilic before fabricating thread-based devices. Sodium carbonate solution was used to remove the hydrophobic substances in threads. Specifically, the cotton threads were firstly soaked in $10 \text{ g L}^{-1} \text{ Na}_2\text{CO}_3$ solution at 100°C for 1 h, then the cotton threads were washed with water until sodium carbonate in the threads was removed. The thread-based device was fabricated with micropipette tips, a knife, a foamed sheet and 0.4-mm diameter cotton threads as described in reference [24]. Briefly, slits were made on the tips followed by inserting into a foamed sheet. The thread was fixed onto the tips by fixing two ends of the thread into tip slits (Figure 1).

Fig. 1 Schematic diagram of a thread-based analytical device.

3.3 Procedure for Tea Polyphenols Assay

$5.0 \mu\text{L}$ of ferrous sulphate solution was pipetted onto the cotton thread with a pipettor, which allow the liquid to wick along the thread. The thread was then allowed to air dry for 2.5 min followed by dropping $7.5 \mu\text{L}$ of tea polyphenols solution onto the thread. After being air dried for 8 min, a blue-black band was formed on the thread. The length of the blue-black band was then measured for quantitative analysis of tea polyphenols.

4. Results and Discussion

4.1 Principle

The length-based detection technique on threads is usually relied on the adsorption or precipitation of colored reaction product on threads [25]. In this work, the hydrophilic cotton threads were firstly modified with chromogenic reagent (ferrous sulphate solution) by dropping chromogenic reagents onto threads (Figure 2a). As sample or standard solutions containing tea polyphenols were spotted onto the thread which was previously modified with the chromogenic reagents, a colorless ferrous-tea polyphenols complex on the thread was formed due to the reaction occurred between ferrous ions and tea polyphenols (Figure 2b). The complex was adsorbed and deposited onto thread fibers until all the analytes were consumed.

As the devices was allowed to air dry, the colorless ferrous-polyphenols could be oxidized to produce blue-black ferric-polyphenols complex. Thus, a blue-black band was formed on the cotton thread (Figure 2c). After the linear correlation between the band length and contents of tea polyphenols was plotted, contents of tea polyphenols in sample were quantified by measurement of the band length resulted from the sample solution.

Fig. 2 (a) Schematic diagrams illustrating deposition of chromogenic reagents (a), formation of ferrous-polyphenols complex (b) and formation of blue-black bands (c) on cotton threads.

4.2 Selection of Chromogenic Reagents

Ferrous tartrate [6,26] and Folin-Ciocalteu phenol reagent [26,27] are the most commonly used chromogenic reagents for determination of polyphenols. Both ferrous tartrate-based and Folin-ciocalteu-based spectrophotometric methods have been used as national standards for detection of polyphenols in tea leaf or tea leaf extracts. Both of these two chromogenic reagents could react with tea polyphenols to produce colored complex on cotton threads. However, the boundary of colored band on the cotton thread is not clear (Figure 3), which make measurement of band length difficult. Furthermore, the band length varied little with the contents of tea polyphenols when using ferrous tartrate or Felin-ciocalteu phenol reagent as chromogenic reagents. These issues may be due to the fact that the colored product could not be adsorbed by the thread fiber. Thus, both ferrous tartrate and Folin-ciocalteu phenol reagent are not suitable as chromogenic reagents for the length-based detection of tea polyphenols on cotton threads.

Fig. 3 Image obtained after adding tea polyphenols solution onto cotton thread modified with ferrous tartrate. Concentration of tea polyphenols: 0.4 g L^{-1} ; Concentration of Fe^{2+} : 0.4 g L^{-1} ; Concentration of tartrate: 6.2 g L^{-1} ; Volume of Fe^{2+} solution: $5.0 \mu\text{L}$; Volume of tea polyphenols solution: $7.5 \mu\text{L}$.

4.3 Effect of Fe^{2+} Concentration

$0.2\text{-}0.8 \text{ g L}^{-1} \text{ Fe}^{2+}$ was prepared for study the effect of chromogenic reagent concentration on the length and intensity of colored band (tea polyphenols concentration was kept constant at 0.4 g L^{-1}). As the concentration of Fe^{2+} increased, the intensity of colored band increased while the blue-black band length decreased with the concentration of Fe^{2+} (Figure 4). $0.4 \text{ g L}^{-1} \text{ Fe}^{2+}$ was used in this work by compromising the color intensity and length of blue-black band formed on the cotton threads.

Fig. 4 The length of blue-black band obtained by varying concentration of Fe²⁺. The error bars were obtained from three replica runs. Concentration of tea polyphenols: 0.4 g L⁻¹; pH of Fe²⁺ solution: 4.5; Volume of Fe²⁺ solution: 5.0 μL; Volume of tea polyphenols solution: 7.5 μL.

4.4 Effect of pH

The effect of pH of chromogenic reagents in the range of 1.0-6.0 on the formation of colored band was studied by keeping the concentration of Fe²⁺ and tea polyphenols constant at 0.4 g L⁻¹ (Figure 5). The results indicated that Fe²⁺ and tea polyphenols could not react to generate colored complex on the cotton threads with ferrous sulphate solution of pH 1.0. The intensity of colored band increased with pH, while the band length decreased with pH in the range of 2-3. Meanwhile, the reaction time needed for formation of colored band decreased from 15 min to 7 min when pH was increased from 2 to 3. Both of the intensity and band length varied little with pH in the range of 3-5. At a pH higher than 5.5, precipitates form in the chromogenic reagent solutions, which may be due to the hydrolysis of ferrous ions. Therefore, pH in the range of 3-5 is suitable for analytical applications by compromising the color intensity and reaction time. In this work, sulphate solution of pH 4.5 was used for analytical applications, which were prepared by dissolving ferrous sulphate in 1:1 ethanol/water without addition of any acid or base solution.

Fig. 5 The length of colored band varying with pH of chromogenic reagent solution. The error bars were obtained from three replica runs. Concentration of Fe²⁺: 0.4 g L⁻¹. Other conditions were the same as in Figure 4.

4.5 Effect of Foreign Species

The effect of main constituents contained in tea leaves on determination of 0.4 g L⁻¹ tea polyphenols was studied, including sugar, ascorbic acid, proteins, amino acids and caffeine. To study the effect of these constituents on determination of tea polyphenols, two solutions were prepared. One contains 0.4 g L⁻¹ tea polyphenols and the other contains 0.4 g L⁻¹ tea polyphenols as well as a foreign species with certain contents. The two solutions were added onto individual threads modified with chromogenic reagents, the length of blue-black band generated by the two solutions was then measured and compared to evaluate the interference of foreign species. The tolerance level was defined as the concentration of foreign species that caused a length variation of lower than 5%. The tolerance levels for determination of 0.4 g L⁻¹ tea polyphenols are 0.4, 0.06, 0.4, 0.2 and 0.4 g L⁻¹ for sugar, ascorbic acid, protein, amino acid and caffeine, respectively.

4.6 Linearity Range and Limits of Detection for Tea Polyphenols

Tea polyphenols solutions with varying concentrations (0.1-0.8 g L⁻¹) was prepared to plot a linear correlation under the optimal conditions described above. The linear correlation was then used to quantify tea polyphenols in samples. After the standard tea polyphenol solutions were added onto threads modified with Fe²⁺, the colored bands were formed (Figure 6a) and the length of these bands were measured. The linear relation between the colored band length (L , mm) and tea polyphenols concentration (C , g L⁻¹) was $L(\text{mm}) = 22.6 C(\text{g L}^{-1}) + 12.2$ (Figure 6b). The correlation coefficient of this curve is 0.994. To obtain the detection limit of tea polyphenols using this proposed method, a solution containing 0.05 g L⁻¹ tea polyphenols was measured 11 times. The standard

deviation was calculated as 0.7 mm. The limit of detection of tea polyphenols is thus 0.09 g L^{-1} according to the equation of $3S/K$.

Fig. 6 The Effect of tea polyphenols concentrations on length of blue-black band. (a) An Image of colored bands formed on cotton threads with varying concentration of tea polyphenols. (b) Calibration curve obtained by varying tea polyphenols concentrations in range of $0.1\text{--}0.8 \text{ g L}^{-1}$. The error bars were obtained from three replica runs. Concentration of Fe^{2+} : 0.4 g L^{-1} . Other conditions were the same as in Figure 4.

4.7 Real Sample Analysis

Polyphenols in tea leaf extracts was detected with this proposed thread-based device. 1.0 g of dried tea leaves was accurately weighed into a 100-mL beaker, followed by pouring 70 mL of boiling water into the beaker. A boiling water bath was used to extract polyphenols in tea leaves for 20 min . The mixtures were filtered to obtain a filtrate. This filtrate was diluted to 100 mL , which was used as a sample. The sample was further diluted by adding $250 \mu\text{L}$ of sample into a 1.5-mL centrifuge tube followed by diluting to 1.0 mL . Detection of tea polyphenols in the diluted sample was then performed on the thread-based devices as described above. The average length of the blue-black band is $21.7 \pm 1.5 \text{ mm}$ ($n=3$). The content of tea polyphenols was $0.42 \pm 0.06 \text{ g L}^{-1}$ in the diluted tea leaf extracts. Therefore, the content in the tea leaf extracts was calculated as $1.68 \pm 0.24 \text{ g L}^{-1}$. Considering that the content of polyphenols is usually much higher than other common constituents in tea leaf extracts, the diluted sample was analyzed without any further pretreatment. The accuracy of this proposed method was evaluated with a ferrous tartrate-based standard method.⁶ The measured contents of polyphenols in the tea leaf extracts is $1.80 \pm 0.04 \text{ g L}^{-1}$, which compared well with that obtained by this proposed thread-based method ($1.68 \pm 0.24 \text{ g L}^{-1}$).

5. Conclusion

We developed a facile method for determination of tea polyphenols in tea leaf extracts on cotton threads with minimum cost. Quantitative analysis of tea polyphenols was achieved by measurement of band length formed on the threads. This simple, portable and cost-effective method could be used to detect tea polyphenols with rapid analysis speed. Furthermore, any electronic instruments was not required during quantitative analysis. We believe that these features make this method very promising and suitable for tea leaf quality control and monitoring especially in those resource-limited setting and regions.

Funding Statement

This research was funded by the Guangdong Provincial Education Research Innovation Fund (No. 2018GXJK108), Guangdong Provincial Science and Technology Innovation Fund for Undergraduates (pdjh2019b0314) and Guangdong Provincial Higher Education Research and Reform Fund.

Data Accessibility

The data supporting the results in this article can be accessed at the Dryad Digital Repository.

Competing Interests

We have no competing interests.

Authors' Contributions

Jiahong Song and Zhuang Ouyang fabricated the thread-based devices and used them to detect polyphenols. Wei Lu evaluate the accuracy of the analytical results using a standard method. Longfei Cai supervised the experiment and wrote the paper.

References

1. Qadir M, T. Muhammad T, Bakri M, Gao F. 2018. Determination of total polyphenols in tea by a flow injection-fiber optic spectrophotometric system *Instrum Sci. Technol.* **46**, 185-193. (doi: 10.1080/10739149.2017.1346515)
2. Zhang SF. 2008. Determination of Polyphenols in Tea by a New Method Spectrophotometry. *Spectrosc. Spect. Anal.* **28**, 1630-1632. (In Chinese)
3. Lau QL, Luk SF, Huang HL. 1989. Spectrophotometric Determination of Tannins in Tea and Beer Samples With Iron (III) and 1,10-Phenanthroline as Reagents. *Analyst* **114**, 631-633. (doi: 10.1039/AN9891400631)
4. Wang JD, Tian LQ, Wang LS. 1995. Thw Application of Atomic Absorption Spectrometry to Organic Analysis. 3. Determination of Tannins in Tea. *Chem. J. Chin. Univ.-Chin.* **16**, 536-539. (In Chinese)
5. Wang L, Yan T, Zhang KX, Li FF, Jia JM, Hu GS. 2019. A sensitive UPLC-MS/MS method for simultaneous determination of polyphenols and theaflavins in rat plasma: Application to a pharmacokinetic study of Da Hong Pao tea. *Biomed. Chromatogr.* **33**, e4470. 1-12. (doi: 10.1002/bmc.4470)
6. Zhou WL, Sun AH, Zhong L. Tea-Determination of tea polyphenols content. Chinese Standard Press: Beijing, China, 2002.
7. Martinez AW, Phillips ST, Butte MJ, Whitesides GM. 2007. Patterned Paper as a Platform for Inexpensive, Low-Volume, Portable Bioassays. *Angew. Chem. Int. Ed.* **46**, 1318-1320. (doi: 10.1002/anie.200603817)
8. Hossain SMZ, Luckham RE, McFadden MJ, Brennan JD. 2009. Reagentless Bidirectional Lateral Flow Bioactive Paper Sensors for Detection of Pesticides in Beverage and Food Samples. *Anal. Chem.* **81**, 9055-9064. (doi: 10.1021/ac901714h)
9. Nouanthavong S, Nacapricha D, Henry CS, Sameenoi Y. 2016. Pesticide analysis using nanoceria-coated paper-based devices as a detection platform. *Analyst*, **141**, 1837-1846. (doi: 10.1039/c5an02403j)
10. Sameenoi Y, Panymeesamer P, Supalakorn N, Koehler K, Chailapakul O, Henry CS, and J. Volckens J. 2013. Microfluidic Paper-Based Analytical Device for Aerosol Oxidative Activity. *Environ. Sci. Technol.* **47**, 932-940. (doi: 10.1021/es304662w)
11. Mentele MM, Cunningham J, Koehler K, Volckens J, Henry CS. 2012. Microfluidic Paper-Based Analytical Device for Particulate Metals. *Anal. Chem.* **84**, 4474-4480. (doi: 10.1021/ac300309c)
12. Mu X, Zhang L, Chang SY, Cui W, Zheng Z. 2014. Multiplex Microfluidic Paper-based Immunoassay for the Diagnosis of Hepatitis C Virus Infection. *Anal. Chem.* **86**, 5338-5344. (doi: 10.1021/ac500247f)
13. Mao X, Du TE, Meng LL, Song TT. 2015. Novel gold nanoparticle trimer reporter probe combined with dry-reagent cotton thread immunoassay device for rapid human ferritin test. *Anal. Chim. Acta* **889**, 172-178. (doi: 10.1016/j.aca.2015.06.031)
14. Hao ZX, Jin LL, Zhou SJ, Yu LZ, Chen HP, Liu X, Lu CY. 2016. Rapid Detection of Tea Polyphenols with Microfluidic Analytical Device. *Anal. Instrum. SI*, 12-13. (In Chinese)
15. Zhou GN, Mao X, Juncker D. 2012. Immunochromatographic Assay on Thread. *Anal. Chem.* **84**, 7736-7743. (doi: 10.1021/ac301082d)
16. Gonzalez A, Gaines M, Gomez FA. 2017. Thread-based microfluidic chips as a platform to assess acetylcholinesterase activity. *Electrophoresis* **38**, 996-1001. (doi: 10.1002/elps.201600476)
17. Sateanchok S, Wangkarn S, Saenjum C, Grudpan K. 2018. A cost-effective assay, for antioxidant using simple cotton thread combining paper based device with mobile phone detection. *Talanta*, **177**, 171-175. (doi: 10.1016/j.talanta.2017.08.073)
18. Cabot JM, Breadmore MC, Paull B. 2018. Thread based electrofluidic platform for direct metabolite analysis in complex samples. *Anal. Chim. Acta* **1000**, 283-292. (doi: 10.1016/j.aca.2017.10.029)
19. Mousavi MPS, Ainla A, Tan EKW, Abd El-Rahman MK, Yoshida Y, Yuan L, Sigurslid HH, Arkan N, Yip MC, Abrahamsson CK, Homer-Vanniasinkam S, Whitesides GM. 2018. Ion sensing with thread-based potentiometric electrodes. *Lab Chip* **18**, 2279-2290. (doi: 10.1039/c8lc00352a)
20. Agustini D, Bergamini MF, Marcolino-Junior LH. 2016. Low cost microfluidic device based on cotton threads for electroanalytical application. *Lab Chip* **16**, 345-352. (doi: 10.1039/c5lc01348h)
21. Lu F, Mao QQ, Wu R, Zhang SH, Du JX, Lv JG. 2015. A siphonage flow and thread-based low-cost platform enables quantitative and sensitive assays. *Lab Chip* **15**, 495-503. (doi: 10.1039/c4lc01248h)
22. Li X, Tian JF, Shen W. 2010. Thread as a Versatile Material for Low-Cost Microfluidic Diagnostics. *ACS Appl. Mater. Interfaces* **2**, 1-6. (doi: 10.1021/am9006148)
23. Reches M, Mirica KA, Dasgupta R, Dickey MD, Butte MJ, Whitesides GM. 2010. Thread as a Matrix for Biomedical Assays. *ACS Appl. Mater. Interfaces* **2**, 1722-1728. (doi: 10.1021/am1002266)
24. Cai LF, Zhang XL, Luo LQ, Lin HB, Chen JH, Xu CX, Zhong MH, Liao XN. 2019. Visual Quantification of Fe on Cotton Thread Using a Ruler. *J. Chem. Educ.* **96**, 1532-1535. (doi: 10.1021/acs.jchemed.8b00800)
25. Nilghaz A, Ballerini DR, Fang XY, Shen W. 2014. Semiquantitative analysis on microfluidic thread-based analytical devices by ruler. *Sens. Actuator B-Chem.* **191**, 586-594. (doi: 10.1016/j.snb.2013.10.023)
26. Turkmen N, Sari F, Velioglu YS. 2006. Effects of extraction solvents on concentration and antioxidant activity of black and black mate tea polyphenols determined by ferrous tartrate and Folin-Ciocalteu methods. *Food Chem.* **99**, 835-841. (doi: 10.1016/j.foodchem.2005.08.034)
27. Obanda M, Owuor PO, Taylor SJ. 1997. Flavanol composition and caffeine content of green leaf as quality potential indicators of Kenyan black teas. *J. Sci. Food Agric.* **74**, 209-215. (doi: 10.1002/(SICI)1097_0010(199706)74:2<209::AID-JSFA789>3.3.CO;2-W)

Appendix B

Responses to the Reviewers' Comments

Reviewer: 1

Comments: *This study presented a novel and cost-effective thread-based method for determination of polyphenols in tea leaf extracts based on length measurements of color bands formed in cotton threads between tea polyphenols and ferrous ions. This analytical method is free of external expensive instruments, which is very interesting and promising for resource-limited regions and settings. I recommend its publication in this journal after fully considering several minor issues as follows.*

Response: We sincerely thank the reviewer for the positive evaluation of this manuscript.

Comment 1: *Tea polyphenols are a class of phenols found in green tea including epigallocatechin gallate (EGCG), epigallocatechin, epicatechin gallate, and epicatechin; flavanols such as kaempferol, quercetin, and myricitin. This study dedicates to total content of tea polyphenols rather than specific or each constitute. Therefore please use total polyphenol content or total content of tea polyphenols all over the text.*

Response: "polyphenol content" was changed to "total polyphenol content" all over the text.

Comment 2: *Colorless ferrous-polyphenol complexes were oxidized to blue-black complexes by oxygen gas in the air or the solvents. I am worrying about variation in the oxidation degree induced naturally. Would you please check if the oxidation was accomplished by using such external oxidants as hydrogen peroxide, sodium hypochlorite, etc.*

Response: the oxidation speed of colorless ferrous-polyphenol complexes on cotton thread is much higher than that happened in centrifuge tube. On cotton threads, color intensity of blue-black complexes increased with time, and the color intensity maintains constant after 6 min, indicating that almost all of the ferrous-polyphenols complexes were oxidized after 6 min on cotton threads. Therefore, it is not necessary to add another oxidant to promote the oxidation process. Furthermore, addition of another oxidant in this assay may pose difficulty in performing assay on threads. Additionally, other oxidant such as hydrogen peroxide may digest polyphenols, causing other interference for the assay.

Comment 3: *In the estimation of interference from coexisting substances, amino acids were considered. As amino acids are a large class of biological molecules, do you test which type of amino acids? If so, please specify the used amino acids.*

Response: glutamic acid was used and we specify it in the text.

Comment 4: *Please provide detailed dimensions of the thread-based analytical device.*

Response: the diameter and length of thread are 0.4 mm and 100 mm respectively. The foamed sheet is 120 mm×120 mm. We provided the dimensions in the revised manuscript (**3.2 Fabrication of Thread-Based Analytical Device** part).

Comment 5: *The description regarding the pretreatment of tea leaves should be placed in the experimental supporting by literature.*

Response: the description of regarding the pretreatment of tea leaves was place in the

experimental section (Materials and Methods section) as the reviewer suggested.

Comment 6: *The manuscript should be proofread to avoid grammatical errors. Some typical grammatical errors are marked in the attached pdf.*

Response: the grammatical errors marked in the attached pdf were revised as the reviewer suggested.

Reviewer: 2

Comments: *In the manuscript (RSOS-192130), Song et al. developed an instrument-free method for polyphenol assay with cotton threads. Quantitative analysis of tea polyphenols was achieved by measurement of band length formed on the threads. It is simple, portable and cost-effective with rapid analysis speed. The results are interesting and it should be of broad interest to the readership of RSOS. But some parts of the text need more detailed explanation and correction. So I recommend this manuscript for publication after minor revisions towards the following points.*

Response: we thank the reviewer for the positive evaluation of our manuscript.

Comment 1: *Ferrous sulphate is soluble in aqueous liquid, why did the authors prepare ferrous sulphate solutions with 1:1 ethanol/water?*

Response: ferrous sulphate solutions was prepared with 1:1 ethanol/water for two reasons: (1) the wicking speed and distance of chromogenic reagents was enhanced. (2) the time needed for air drying was decreased from 10 min to 2.5 min, which could improve the assay speed.

Comment 2: *Why the author does not use the untreated cotton threads to fabricate the device? Are there any other method or solution could be used to pretreat the hydrophobic threads besides Na₂CO₃ solution?*

Response: the treated (hydrophobic) cotton threads are more available in the local market. Additionally, the treatment procedure is not difficult and large amount of cotton threads could be treated in batch. Besides Na₂CO₃ solution, NaOH and NaOH-SDS solution could also be used to pretreat the hydrophobic threads.

Comment 3: *In the manuscript, author claimed that the limit of detection of tea polyphenols is thus 0.09 g L⁻¹. However in the text, "a solution containing 0.05 g L⁻¹ tea polyphenols was measured 11 times ..." How can such a low concentration of signals be detected?*

Response: usually, the detection limit was calculated according to the equation of $3S/K$, where S is the standard deviation obtained from 11 repetitive assays of blank or standard solution with low concentration of analytes. The distance signal could also be detected when 0.05 g L⁻¹ tea polyphenols was used. A detection limit higher than 0.05 g L⁻¹ tea polyphenols was obtained, this may be due the high standard deviation calculated from 11 repetitive assays.

Comment 4: *Could distance-based detection of polyphenols be performed and achieved on paper-based analytical device using reaction between ferrous sulphate and polyphenols?*

Response: we fabricated paper-based analytical device with a straight paper channel delimited by hydrophobic wax barrier, which was used as a platform to perform distance-based detection of tea polyphenols. However, the length of colored bands varied little with the tea polyphenols concentrations on paper devices.